# Subjective Perception Types of Older Adults Living Alone as Farmers in Korea: A Q Methodology Study

**DOI:** 10.3390/bs14121150

**Published:** 2024-12-01

**Authors:** Oh Sun Ha, Song Yi Lee

**Affiliations:** 1Academic Affairs Team, Dongguk University-Seoul, Seoul 04620, Republic of Korea; haosun@dongguk.edu; 2Department of Counseling and Coaching, Dongguk University-Seoul, Seoul 04620, Republic of Korea

**Keywords:** older adults, farmers, living alone, quality of life, relationship, family

## Abstract

This study explores the psychological characteristics of older adults living alone as farmers in South Korea, categorising their subjective experiences. Using Q methodology, interviews were conducted with participants from Seosan, Chungcheong Province, South Korea, on 19 and 22 June 2023. A total of 25 participants conducted Q sorting with 33 finalized Q sample items in three locations: Chungju, Chungcheong Province, on 14 July 2023; Ildong, Gyeonggi Province, on 28 July 2023; and Seosan, Chungcheong Province, on 14 August 2023. The Q sorting results were analysed using the QUANAL program and categorised into four types. Type 1, Balanced Life Pursuit, reflects satisfaction across various aspects of life, such as relationships, health, economy, and psychological wellbeing. Type 2, Independent Life and Improvement-Seeking, pursues independence but experiences loneliness due to the absence of a spouse, while still seeking to improve their life. Type 3, Relationship and Care Needed, highlights a need for relationships and care. Finally, Type 4, Family and Work-Focused, prioritises family and farming work. These findings provide a foundation for developing customised support programs to address the specific needs of different types of older adults living alone as farmers in South Korea.

## 1. Introduction

South Korea is expected to become a super-aged society by 2025, with the population aged 65 and older reaching 20.6% [1]. The ageing population in rural areas is particularly concerning, with 46.8% being 65 or older, and approximately 1.38 million people aged 60 or above, making up 62.4% of the total rural population [2]. To address the social and economic issues arising from this increase in the older adult population, the government has been conducting surveys every three years since 2008, and has implemented various initiatives, including welfare policies, customised services, welfare facilities, infrastructure, and social participation programs for older adults [3].

The 2020 survey by the Ministry of Health and Welfare [3] found that 19.8% of older adults live alone, with women making up 74.6% of these households—three times more than male households (25.4%). Older adults living alone rate their happiness lower than their married peers or those living with partners [4]. They also have fewer opportunities for support compared to older adult couples or those living with their children, and experience a weakened social support network [5].

The duration of living alone is associated with psychosocial issues: the more time spent living alone, the stronger the thoughts of suicide. These individuals also experience vulnerabilities in economic status, health, and relationships [6,7]. Moreover, there is a positive correlation between self-esteem and health-promoting behaviours among older adults living alone; lower perceived health status is associated with higher levels of depression, indicating that psychological characteristics affect the quality of life of older adults living alone [8,9,10].

The depression rate among older adults with functional limitations is 41.8%, more than four times higher than the 10.0% rate among those without functional limitations [3]. About 80% of older adults living alone experience difficulties in daily life, highlighting the importance of functional ability [5]. Risk factors for functional decline include older age, being female, low economic satisfaction, low social activity, and joint disorders. In particular, functional decline reduces social activity [11]. Jeon and Zheng [12] found that participation in job programs increased life satisfaction, particularly for older adults. Social activity through work also positively impacts older adults’ health [13].

Depression has a higher impact on suicidal thoughts among older adults living alone in rural areas than those in urban areas [14]. This is due to a lack of access to medical services that address psychological issues, increasing the likelihood of depression leading to suicidal thoughts [14]. The prevalence of suicidal thoughts among older adults living alone in rural China is 23.6%, with depression being the most significant factor, while physical and economic status have indirect effects [15]. A literature review identified six risk factors for suicide among older adults in rural areas: health challenges, unmet basic needs, abuse from children, loneliness, negative life events, and altruistic motivation to benefit children [16].

Compared to urban counterparts, rural older adults face challenges, such as residing in single-family houses (77.9%), which often require maintenance that they cannot afford [17,18]. Older adults are a heterogeneous group with diverse needs, and as they age, they require housing environments that adapt to their changing abilities [19].

Leisure activity satisfaction among rural older adults is key to enhancing psychological wellbeing [20]. Lee et al. [21] confirmed that participation in leisure activities is significantly associated with positive emotions, an important component of healthy ageing [22]. For older women living alone, communal living reduces the negative effects of declining daily activities [23].

A study using Q methodology categorised female older adults living alone into three types: overcoming reality and self-reliance, denial of reality and resignation, and acceptance of reality and conformity [24]. Research on depression in this group showed that after being left alone, they struggled through harsh conditions, unable to cope with loneliness and suffering [25].

Although the overall economic and health status of older adults is improving, those living alone as farmers remain particularly vulnerable. They face isolation and lack of social contacts, necessitating specialised policies and attention [24]. Conway et al. [26] emphasised the urgent need to re-evaluate South Korea’s existing approach, which prioritises younger farmers, advocating for equal emphasis on the quality of life of older adults living as farmers.

Despite various forms of support, older adults’ health, psychological, economic, and family statuses vary. Ageing does not necessarily worsen life perceptions, but subjective wellbeing influences life satisfaction [27]. The psychological characteristics of older adults living alone as farmers are complex, intertwined with family, health status, farming, productivity, and economic satisfaction. However, existing measures to address these characteristics are insufficient.

Uniform support programs have limitations for older adults living alone as farmers. Understanding their subjective experiences is crucial for developing better support. This study characterises these experiences to help create tailored services, moving away from one-size-fits-all programs.

## 2. Materials and Methods

This study uses the Q methodology to explore the psychological characteristics of older adults living alone as farmers. William Stephenson [28] developed this methodology in 1935 to focus on the human mind’s internal attributes, moving away from universally objective measurements and exploring personal diversity. The Q methodology comprises the quantitative research aspect of typifying and the qualitative feature of interpreting each type’s characteristics.

Regarding qualitative attributes, this methodology is an innovative approach to conducting qualitative research by quantifying subjective tendencies or values held by individuals and analysing the interpreted value types [29]. McKeown and Thomas [30] highlighted the complementary nature of Q methodology’s quantitative and qualitative research approaches to explore subjectivity, incorporating factor analysis. However, Q findings aim to develop theories, similar to qualitative research. Instead of statistical inference about a population, Q researchers may emphasise substantive inference, linking broader generalisations about phenomena to a population [31].

This method is beneficial when analysing less structured concepts such as personal views, preferences, and images, which can elicit more diverse responses than highly structured perceptions such as ideology, beliefs, knowledge, and facts. In other words, the Q methodology provides a foundation for examining individuals’ subjective perspectives, views, opinions, beliefs, and attitudes towards specific subjects or situational contexts. Because this study evaluates the subjective perspectives of older adults living alone as farmers regarding their experiences, the Q methodology is the ideal research method.

The research procedure involves the following process (Figure 1): building the Q population, selecting the Q sample, selecting the P sample, Q sorting, and analysis.

### 2.1. Q Population Construction

The Q population comprises items related to the research topic, representing the totality of perceptions shared within a specific culture. Researchers can use various methods to build the Q population, including questionnaires, literature reviews, and in-depth interviews—this study conducted a literature review and in-depth interviews and presented items as statements.

Using the Research Information Sharing Service (RISS) in Korea, we conducted an electronic search of the literature using keywords such as ‘older adults living alone as farmers’, ‘older adults living alone’, ‘elderly’, and ‘rural areas’. Based on these keyword searches, we selected and analysed relevant journals and papers [3,20,23,24,25,32,33,34,35,36,37,38]. We extracted 259 statements from these materials about various aspects of older adults living alone as farmers, including their relationships with family and neighbours, health awareness and management, rural life, and attitudes towards retirement and ageing. For example, D.J. Kim [32] extracted checklist items related to quality of life and attitudes towards life, H. Kim [33] focused on thoughts and feelings about oneself and relationships with others, while Choi and Kim [36] highlighted preparations for retirement life.

In formulating these statements from the literature, we designed the content to reflect the perceptions of older adults living alone as farmers, adhering to H. Kim’s [33] statement composition criteria. First, every statement must be relevant to the research topic. Second, every statement must be self-referential, allowing for subjective agreement or disagreement with each statement. Third, the statements must reflect the level of the study participants. Specifically, the guidelines suggest using conversational language—in this study, the statements are simple, considering the education level and comprehension of older adults living alone as farmers. Fourth, each statement should contain one clear idea. We followed the same criteria when formulating statements during the in-depth interviews with older adults living alone as farmers.

We interviewed seven seniors living alone using a semi-structured questionnaire. The interview questions included: “What does living alone mean to you”, “What are the difficulties of living alone”, “What are the benefits of living alone”, “How satisfied are you with your current life and why”, “What are your thoughts on family and why”, “What does farming mean to you”, and “Can you describe the difficulties or benefits related to farming”? This study used purposive sampling, with participants selected by a community contribution officer from the National Agricultural Cooperative Federation.

The interviews took place on 19 June 2023 in Seosan, Chungcheong Province, and on 22 June 2023 in Hapcheon, Gyeongsang Province. As a token of appreciation, we compensated each interviewee KRW 50,000. Three professors conducted the in-depth interviews: one with a PhD in psychology, one with a PhD in education, and one with a social worker certification in psychology. All of them were familiar with the Q methodology and had completed ethics training. Additionally, they were part of the research team for this study.

With the participants’ permission, the researchers recorded and transcribed the interviews; each interview lasted about one to two hours. The researchers reviewed the transcripts and recordings together for a comprehensive understanding, extracting 148 statements that reflect the psychological characteristics of older adults living alone as farmers. Ultimately, the Q population comprised 407 statements, with 259 extracted from the literature review and 148 from interviews.

Table 1 presents the demographic background of this cohort.

### 2.2. Q Sample

The Q sample represents a subset of the Q population that can comprehensively cover the topic. Each item in the Q sample must be relevant to the research topic and self-referential [39]. The Q sample represents a subset of the Q population that can comprehensively cover the topic. When selecting the Q sample from the Q population, there is no need to base the selection on theory; instead, the researcher should carefully classify and construct the sample to meet the criteria of representativeness [40]. The Q sample statements should not measure specific variables or constructs; instead, they should capture various perspectives about the research topic [41]. Baek [42] emphasizes that researchers should avoid biasing the Q sample towards any particular viewpoint or opinion. The selected items should collectively cover the entire research topic while holding individual significance contributing to the whole.

To select a balanced Q sample in this study, researchers repeatedly reviewed the 407 items, excluding those deemed irrelevant to older adults living alone as farmers, merging items with similar meanings, and removing duplicates. This process resulted in categories of health, relationships (family, neighbours, etc.), farming, perceptions of life, self-perception, use of government support, economic satisfaction, daily life, contributions to children, living environment, independence, and hobbies. We selected statements according to their proportions in these categories. In the first meeting, researchers derived 152 items; in the second, 95 items; and in the third, 40 items. Finally, considering the age and educational level of the P sample, the researchers selected 33 Q sample statements, as detailed in Table 2.

### 2.3. P Sample Selection

The P sample refers to the participants conducting the Q sorting. Q studies typically involve small numbers of participants [43]. Furthermore, since the aim is to generate types, the sample size only needs to be sufficient for meaningful comparison and should be relevant to the research topic [44]. According to Paik and Kim [45], P samples can range from 10 to 100 participants. More often than not, Q studies use non-random sampling, such as purposive sampling [43]. The focus is not on generalising the research results but on identifying patterns of subjective thoughts within the Q population, believing that these patterns exist in the population represented by the sample [46].

This study’s P sample comprised 25 older adults living alone as farmers in rural areas, recruited following purposive sampling and ensuring participants represented various age groups, genders, and regions. However, the sample was predominantly female due to the significantly higher number of female older adults than male counterparts [3]. Additionally, we limited the study to regions where participants expressed willingness to participate, focusing on Ildong, Gyeonggi Province; Chungju, Chungcheong Province; and Seosan, Chungcheong Province. The characteristics of the P sample are displayed in Table 3.

### 2.4. Q Sorting

In Q sorting, participants assign scores to each item. This study included older adults living alone as farmers, who ranked the 33 Q sample items based on their subjective perceptions. As shown in Figure 2, participants sorted cards with each statement using a Q-grid (Figure 3). Using a sheet for Q sorting (Figure 4), participants wrote their reasons for selecting the statements with which they most agreed and disagreed. Six psychology master’s students, three psychology PhD students, and two authors of this study guided participants in Q sorting. The sessions took place in Chungju, Chungcheong Province, on 14 July 2023, Ildong, Gyeonggi Province, on 28 July 2023, and Seosan, Chungcheong Province, on 14 August 2023.

Before the fieldwork, the researchers attended an eight-hour training session on Q methodology. They also received training on Q sorting from two university professors specializing in Q methodology. Considering the demographic characteristics of the P sample, the researchers printed the statements in a larger font for clarity. A facilitator assisted each participant throughout the Q sorting process to ensure smooth progress.

The researchers conducted 1:1 Q sorting with the P sample using a 33-item Q sorting grid, as demonstrated in Figure 2. The Q sorting procedure followed the steps outlined by Kim [39]. First, we explained the study’s purpose and the sorting method to the participants. Second, participants (P sample) read the statements on the cards and broadly classified them into three groups: positive, neutral, and negative. Next, participants selected the statements with which they agreed and ordered them from the “most disagreeable” to a neutral position to the “most agreeable” (+) categories. Fourth, they selected the statements with which they disagreed and organized them from the “most agreeable” to neutral to “most disagreeable” (−). Fifth, participants reviewed their Q sort for accuracy and made desired changes. Finally, for the cards placed at the extremes of the distribution, participants explain, in written form or through an interview, why they strongly agreed or disagreed with the statements.

This process occurred at community centres in Chungju, Chungcheong Province, Ildong, Gyeonggi Province, and Seosan, Chungcheong Province, with each Q sorting session lasting between one hour and one hour and 20 min per participant. While the Q sorting in Huang et al.’s [47] study took approximately 50 min, this study required a slightly longer time.

### 2.5. Data Analysis

The researchers analysed the Q sorts using the QUANAL program, developed by Norman Van Tubergen in the 1960s, for mainframe platforms. We determined the number of factors by entering numbers ranging from 2 to 10 into QUANAL and selected the results with the highest explanatory powers. The advantage of QUANAL is that it can produce many results with simple data input [48]. We used Q principal component factor analysis to analyse and interpret the resulting data (Z-scores), exploring the significance of these items. Principal component analysis extracts factors to maximise the explanatory power of variables, offering greater explanatory power than centroid analysis [39].

This study used principal component analysis with atheoretical rotation and the varimax method to maximise factor rotation. The results provide the explanatory power, factor correlations, Z-scores for each item per factor, factor weights, and common items. Principal component analysis typically extracts only factors with eigenvalues greater than 1.00. We loaded Z-scores onto all items to analyse their significance. Moreover, the difference highlights the significance of a particular item in a specific type compared to others. To interpret this study’s results, we focused on the items with Z-scores and differences of 1.000 or more and referred to the perception of P samples with high factor weights for each type. A P sample with a high factor weight can represent each factor’s characteristics [39]. Additionally, we analysed the P sample’s detailed, written reasons for identifying certain items as the most agreeable.

Before the Q sorting process, the researcher explained each concept to the participants to ensure they understood the Q items. However, participants may have different reasons for developing certain competencies, which explains why they strongly agreed with some items, reflecting their subjective perceptions. Therefore, we identified and interpreted each type based on factor analysis and the descriptions provided in the Q-sort sheet. We then extracted the related types using factor statements, including the P sample’s most agreeable and most disagreeable selections.

## 3. Results

The study identified four factors representing the psychological characteristics of older adults living alone as farmers. As shown in Table 4, the eigenvalues for each type were as follows—Factor 1 = 5.2856, Factor 2 = 3.3980, Factor 3 = 2.0356, and Factor 4 = 1.6975, with a cumulative variance of 0.4967.

The correlations between individual types, as detailed in Table 5, indicate the degrees of similarity between them. The correlation between Types 1 and 2 was 0.131, Types 1 and 3 was 0.090, and n Types 1 and 4 was 0.470. The correlation between Types 2 and 3 was 0.172, Types 2 and 4 was 0.262, and Types 3 and 4 was 0.039.

### 3.1. Type 1: Balanced Life Pursuit

Table 6 presents the statements with which Type 1 agreed and disagreed. The statements with which the five participants in Type 1 agreed most are: “I believe my family understands me” (Q17, Z = 2.00), “I spend a lot of time watching television, which entertains me, so I feel like it is my good friend” (Q23, Z = 1.84), “I wish my family would take care of me” (Q2, Z = 1.62), “Although I am struggling with chronic conditions (high blood pressure, diabetes, arthritis, etc.), I am still trying to stay healthy” (Q1, Z = 1.31), “I have never thought about doing anything other than farming, and I want to farm for the rest of my life” (Q15, Z = 1.21), “I still have an appetite, and although I live alone, I eat well” (Q3, Z = 1.19), and “I am happy when my family visits often and helps with my household chores and farming” (Q16, Z = 1.09). However, Type 1 disagreed with “Being alone makes me feel lonely and empty” (Q25, Z = −1.63) and “I think my health is worse than others who are in my age range” (Q4, Z = −1.46).

The following insights emerged from the statements and reasons given during the Q-sorting interviews by participants with high factor weights. For example, P1 said, “If I stay at home, I only get sicker, and doing nothing makes me worse. Although my children tell me not to, I do it for exercise because I have a field”. P16 said, “Farming makes me healthier”, and P9 stated, “Although I live alone, I am not lonely. I keep myself busy, and such thoughts don’t cross my mind. At my age, I should keep working until I can’t anymore”. These comments demonstrate these participants’ strong willingness to continue farming. In addition, P16 said, “Cooking and eating well is important for my health, so I do it willingly”.

Type 1 finds joy and satisfaction in farming and family connections, as demonstrated by the following statements: “Farming is my joy because I can give my children and grandchildren what I’ve grown myself. That brings me happiness” (P16), “Being alone, my son tells me I must live a long life. I feel cared for and cherished by my family” (P1), and “My grandson visited yesterday, asked how I was, and ate with me. He told me not to get sick, and I find my grandson adorable” (P20). Type 1 seniors also appreciate their neighbours’ small acts of kindness, reporting, for instance, that “Sometimes neighbours come over to heat food for me and help a lot. They even turn on the lights in advance, so it won’t be dark when I come home” (P16). These older adults living alone as farmers feel gratitude and joy in their relationships with family and neighbours, finding satisfaction and balance in their daily lives.

Based on these findings, we named Type 1 “Balanced Life Pursuit”. These older adults living alone as farmers do not feel lonely and believe they are healthier than their peers. They take good care of their meals and have positive family relationships, feeling genuinely understood by their family members. They also actively engage in farming. They are satisfied in various aspects of life, including relationships, health management, farming, finances, and overall life, demonstrating a well-balanced life.

### 3.2. Type 2: Independent Life and Improvement-Seeking

The statements with which Type 2 agreed and disagreed are in Table 7. Twelve older adults living alone as farmers fall under Type 2, and they strongly agreed with statements such as “Even if my children help and try to make me feel happier, it will never be the same as when I lived with my spouse” (Q19, Z = 1.81), “I want to fix my house (heating system, bathroom, rat infestation, etc.) because it is uncomfortable to live in” (Q30, Z = 1.18), “I am grateful that I can live like this and am satisfied with my current life” (Q26, Z = 1.04), and “I believe that when going through hard times, I should not expect others to help me and should solve problems by myself” (Q28, Z = 1.00). However, they disagreed with statements like “I am confident in my ability to farm and handle farming equipment” (Q14, Z = −2.24), “I do not need a lot of money, and I am satisfied because I have enough money to do what I want” (Q8, Z = −1.83), “I think my life has been quite successful, and I have achieved a lot of what I wanted” (Q27, Z = −1.10), and “I wish my family would take care of me” (Q2, Z = −1.05).

The following insights, drawn from participants’ statements during the Q-sorting interviews, provide a clearer understanding of Type 2’s characteristics, focusing on those with high factor loadings. For instance, P4 said:

It’s very lonely and desolate without my husband (who died of illness 5 years ago). My son used to handle the farm equipment, but he died in an accident, and now it’s very sad and lonely. Living alone is financially difficult. My husband is gone, my son is gone, and my daughter lives in Seoul and only visits once a year.

P15 reported, “Everyone feels this way. I used to rely on my husband, but he is gone”, and P11 said, “A husband (spouse) is better than children. Living alone is lonely and desolate”. A characteristic of this type is extreme loneliness, which they attribute to a spouse’s absence; they also used to rely heavily on their spouses for farming tasks.

Nevertheless, they also exhibited a strong sense of independence. For example, P7 said, “I don’t want to burden others, and it’s better to solve my problems by myself. In addition, P12 stated, “In every aspect, I try hard to live without relying on others”, while P10 said, “I farm blueberries, so I can live well without asking my children for help”. In addition, P12 reported “Taking care of my health and living a long, healthy life is important. Eating well is essential to maintain health”. Participants also expressed their desire for an improved life: “I want to live longer because the world is so wonderful now, my children are all grown up, and I no longer struggle as I did in the past. I enjoy meeting friends, chatting, and engaging in hobbies” (P13), and “I want to rebuild and renovate my house to live more comfortably” (P4).

Based on these results, we labelled Type 2 “Independent Life and Improvement-Seeking”. These seniors feel lonely without their spouses and live far from their children. They also lack confidence in farming because their spouses used to manage the farming tasks, and they no longer have the strength required for such work. Compared to other groups, they are more inclined to stop farming, rest at home, and engage in hobbies. They also desire to renovate their current homes, improve their living conditions, and enjoy a better life. Despite economic difficulties, they consider it fortunate that they can support themselves through farming without relying on their children and strive to solve problems independently, thus reflecting an independent life attitude.

### 3.3. Type 3: Relationship and Care Needed

Type 3 agreed with certain statements and disagreed with others, as demonstrated in Table 8. Three participants strongly agreed with statements such as “I am happy and grateful even if my family visits briefly” (Q18, Z = 1.72), “Being alone makes me feel lonely and empty” (Q25, Z = 1.59), “I wish my family would take care of me” (Q2, Z = 1.54), “My vision, hearing, and dental condition have deteriorated, making me feel uncomfortable and sick, leading to not only physical but also psychological pain” (Q5, Z = 1.52), “Since government agencies call and check on me daily, I think they are better than my children” (Q20, Z = 1.14), and “Although I am struggling with chronic conditions (high blood pressure, diabetes, arthritis, etc.), I am still trying to stay healthy” (Q1, Z = 1.09). They strongly disagreed with statements like “Farming is physically demanding and exhausting, but I want to continue doing it as long as I can” (Q9, Z = −1.89), “I am confident in my ability to farm and handle farming equipment” (Q14, Z = −1.34), and “I still have an appetite and, although I live alone, I make sure to eat well” (Q3, Z = −1.15).

To understand Type 3’s characteristics more deeply, we examined the reasons provided by participants with high factor loadings. P5 said, “I have no particular attachment to farming, and my hearing is poor, making communication difficult. I am psychologically distressed and can no longer farm due to my poor health”. In addition, P6 said:

I am too old to farm, have digestive problems, and find it hard to cook for myself. It would be good if my children took care of me. I hurt my back and can’t work, so a caregiver visits me now. The caregiver doesn’t come on weekends, so I eat alone. The caregiver comes Monday to Friday, but I’m alone on weekends. A friend visits, and we talk about family, local news, and other things. It’s enjoyable to talk.

These responses indicate a strong need for relationships and care.

Based on this analysis, we named Type 3 “Relationship and Care Needed”. Type 3 older adults living alone as farmers feel lonely and suffer psychological pain due to poor health. They lack confidence in farming and have no desire to continue. While they experience less economic hardship, they prefer to rely on family, neighbours, and government care services. Unlike other groups, they struggle to manage meals independently and hope for help from others when they are in need. Thus, this type needs relationships and care due to loneliness and poor health.

### 3.4. Type 4: Family and Work-Focused

Table 9 summarises the statements that Type 4 participants most strongly agreed or disagreed with. The five participants of this type strongly agreed with the following statements: “I feel happy that I can give my children what I have grown myself and buy things for my grandchildren” (Q13, Z = 2.21), “I believe my family understands me” (Q17, Z = 1.71), “I feel satisfied when my crops grow well, but I get upset when farming doesn’t go well due to problems like the weather or wild boars” (Q11, Z = 1.25), and “I am happy and grateful even if my family visits briefly” (Q18, Z = 1.21). However, they disagreed with these statements: “I want to quit farming now and rest at home, or engage in other hobbies” (Q29, Z = −1.50), “My vision, hearing, and dental condition have deteriorated, making me feel uncomfortable and sick, leading to not only physical but also psychological pain” (Q5, Z = −1.33), and “I wish my family would take care of me” (Q2, Z = −1.10).

To understand Type 4’s characteristics more comprehensively, we assessed the statements of Type 4 participants with high factor weights. Among their responses, P2 claimed, “I am grateful that my children care about me, and I feel upset when the crops, which I have nurtured like my own children, get damaged or die”, and P14 said, “I prefer my children over daily calls from government agencies, but my husband is the best. I am only a farmer; I don’t have another job”. These comments indicate that they highly value their family and farming work. Additionally, they displayed a positive attitude towards life, saying, “I try to endure even when I get annoyed, aiming to live happily. I have had thyroid surgery in the past, but I do not feel psychological pain” (P2) and “Although I have high blood pressure, I try to stay healthy and make sure to eat well” (P25).

Based on the above analysis, Type 4 is classified as the “Family and Work-Focused” type. Individuals in this cohort derive happiness from providing homegrown food to their children and purchasing items for their grandchildren. They perceive themselves as physically healthy and feel satisfaction when crops thrive, though they experience frustration when farming faces challenges, such as adverse weather conditions. They are grateful that their family understands them, appreciate having neighbours, and feel happy when their family helps with farming. They also find joy in providing something for their grandchildren through farming. Compared to other groups, they prioritise farming and value their family highly.

### 3.5. Consensus Statements of Individual Types

The statements in Table 10 represent those that are common across all types.

## 4. Discussion

This study categorised older adults living alone as farmers based on their subjective perceptions of life and explored the psychological characteristics of how they perceive their lives. We identified four types: Balanced Life Pursuit (Type 1), Independent Life and Improvement-Seeking Pursuit (Type 2), Need for Relationships and Care (Type 3), and Family and Work-Oriented (Type 4).

Type 1 individuals are generally satisfied with their lives. They consider themselves relatively healthy and strongly disagreed with the statement, “I think my health is worse than that of others my age”. This finding aligns with Song and Son [9] and Kwak and Song [49], who suggest that a positive subjective perception of one’s health correlates with lower levels of depression and higher ratings of health-related quality of life. Type 1 seniors stated, “If I stay idle because I feel sick, it will only make things worse”, and “I do farm work as a form of exercise”, indicating their intention to continue farming activities to maintain physical health. According to Rúa-Alonso et al. [50], engaging in recommended physical activity levels is particularly important. It appears that Type 1, comprising female older adults in our study, considers farm work a moderate level of exercise. This perception contributes to a higher quality of life.

Type 1 individuals expressed feeling cherished by their family and voiced gratitude for the help they receive from their neighbours. They find joy in giving their farm produce to their children. These individuals frequently use positive expressions such as “grateful”, “thankful”, and “happy” when discussing their lives, reflecting a highly positive attitude. Although they live alone, they maintain a satisfying life through their farming efforts, a positive attitude, and emotional bonds with their family, preventing them from feeling lonely. Gubrium and Holstein [51] argued that, from a constructivist perspective, a positive ageing experience needs a repertoire of resources to create primarily positive meanings. Type 1 individuals have abundant social resources through family and neighbours, psychological resources through positive thoughts and a grateful mindset, and active resources through farming work. These resources drive their balanced life.

Type 2 individuals face significant loneliness due to the absence of their spouses. They strive to overcome these challenges by independently improving their environment and solving life’s problems. All members of Type 2 are female older adults living alone. They stated, “A husband (spouse) is better than children. Living alone is lonely and desolate. I used to rely on my husband”. They feel lonely because their husbands are absent and experience difficulties as their farming capabilities, for which they once relied on their husbands, have decreased. This finding supports Kim, Lee, and Park [23], who found that female older adults farming alone in rural areas experience decreased life satisfaction as their instrumental activities of daily living decline.

Nevertheless, they find their children’s visits a source of vitality and strive to be satisfied with their current lives. They believe in overcoming difficulties independently rather than relying on others. Additionally, they expressed a strong will to improve their lives, saying, “I want to meet friends, chat, have hobbies, and build and renovate a new house to live better. I want to live longer because this world is wonderful”. This comment reflects their desire for a better life and aligns with Park [20], who found that leisure activity satisfaction is crucial for enhancing the psychological wellbeing of older adults, particularly those living alone in rural areas. It highlights the need to support seniors’ involvement in hobbies to enhance the psychological wellbeing of Type 2 individuals.

Type 3 individuals do not experience significant economic hardship, such as issues related to retirement preparation. However, they feel lonely and suffer physical and psychological pain due to poor health, indicating their need for relationships and care. According to a 2020 survey administered by South Korea’s Ministry of Health and Welfare [3], older adult individuals living alone with functional limitations in important life activities are four times more likely to exhibit depression symptoms than those without such limitations. Type 3 individuals suffer from poor vision, hearing, and dental conditions, which cause discomfort and pain. They also experience significant limitations in farming and daily activities.

These individuals hope their children or family will care for them but do not necessarily expect it. They rely on friends and neighbours and are satisfied with care services provided by government agencies. Although the family is the primary caregiver for older adults in most societies, the significance of family, caregivers’ roles, and family care expectations can vary depending on the cultural background and personal history of people living in different societies [52]. Kim et al. [48] suggest that in a super-aged society, companion robot services are essential for managing the daily lives of older adults living alone and providing emotional support. This approach could be particularly useful for Type 3 individuals. This study’s participants only included two men, and both were Type 3, indicating that male older adults living alone value emotional and practical care.

Type 4 participants consider themselves healthy despite their old age and remain independent, focusing on family and work. They focus more on farming than the other types and live independently through the produce they grow by farming, feel joy in supporting their children, and express gratitude towards their families. According to Han [53], seniors can enhance their mental wellbeing by being satisfied with their current situation and repeating achievement activities. Members of this demographic exhibit a positive mindset, accepting their situation and striving to stay healthy, as indicated by statements like “I try to endure even when I get annoyed, aiming to live happily” and “Although I have high blood pressure, I try to stay healthy and make sure to eat well”.

They emphasise the importance of farming in their lives and strongly agree with the statement, “I feel happy and pleased that I can give my children what I have grown and buy things for my grandchildren” (Q13, Z = 2.21). This statement shows that they find psychological satisfaction through the repetitive achievement of farming activities. According to Kang [54], a job increases self-esteem and reduces feelings of isolation, and social activities through work positively impact older adults’ health [13]. As people age, their life satisfaction increases with participation in work activities [12], and job satisfaction positively influences overall life satisfaction [55]. Thus, Type 4 individuals are satisfied with their lives, as they find attachment in their communities and achievement in their farming work.

Based on this study’s outcomes, we recommend the following. First, regarding the “Balanced Life Pursuit” (Type 1), local community centres should provide older adults cultural and leisure activities to strengthen their social relationships. Regular health check-ups and exercise programs will help them maintain their health, while continuous agricultural skills training and support programs will help them maintain their farming work satisfaction.

Second, for those classified as “Independent Life and Improvement-Seeking Pursuit” (Type 2), it is necessary to provide programs for new hobbies or leisure activities to promote changes in their lives. Support programs to improve their living environment are essential for enhancing their quality of life. Additionally, financial counselling and economic support programs will ensure their economic stability.

Third, the “Relationship and Care Needed” (Type 3) older adult requires regular care services and home visits to ensure physical and psychological stability. Social interaction programs such as self-help groups and mentoring programs may reduce social isolation and help these seniors form relationships. Additionally, meal support programs providing nutritious food or meal delivery services are essential to maintain a proper diet.

Finally, for adults in the “Family and Work-Focused” (Type 4) group, providing technical support and materials to enhance agricultural productivity is essential. Family participation schemes that involve agricultural activities or family visit programs can strengthen family bonds. Additionally, community activities to promote interaction with neighbours are necessary to support social engagement within the individual’s local community.

## 5. Conclusions

This study classified older adults living alone as farmers in South Korea into four types: Balanced Life Pursuit (Type 1), Independent Life and Improvement-Seeking Pursuit (Type 2), Need for Relationships and Care (Type 3), and Family and Work-Oriented (Type 4). These classifications provide valuable insights into the diverse experiences and needs of these populations, highlighting their psychological, emotional, and practical challenges.

While Types 1 and 4 maintain stability and satisfaction through balanced relationships, health, and farming activities, Types 2 and 3 experience more emotional loneliness and require targeted support. Type 2 seeks improvements for a better living condition, whereas Type 3 is more reliant on family and social services due to physical and psychological distress.

These insights are crucial for developing targeted support programs tailored to each type’s specific needs. Effective interventions could include emotional support, community-building initiatives, and improved healthcare services to address the diverse challenges faced by older adults living alone as farmers.

Additionally, examining the role of government and community support systems could provide a more comprehensive approach to addressing the challenges older adults face in rural areas.

## 6. Limitations

This study identified four subjective perception types among older adults living alone as farmers, providing a foundational resource for developing customized programs. However, the study has several limitations when interpreting these findings.

First, the study’s sample had a low proportion of male participants. Future quantitative research can be done to analyse gender differences in perceptions with a more balanced representation of men and women. Second, we limited the survey to two rural areas in South Korea, which may not capture the full range of cultural differences across regions. Future studies should include participants from a broader geographical range to gain a more comprehensive understanding. In addition, longitudinal studies could help researchers understand how this population’s needs evolve.

Additionally, using Q methodology to assess perception types makes generalising this study’s results challenging. Follow-up quantitative research is necessary to explore these types further and consider the demographic backgrounds that may influence such perceptions. Future research should also focus on developing a scale to identify the quality-of-life characteristics of older adults who live alone as farmers in South Korea. Researchers and policymakers could use a simplified scale to categorise individuals based on their psychological characteristics, and design customized programs to enhance their wellbeing.

## Figures and Tables

**Figure 1 behavsci-14-01150-f001:**
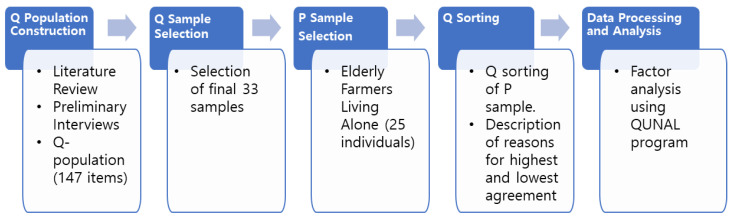
Research procedure.

**Figure 2 behavsci-14-01150-f002:**
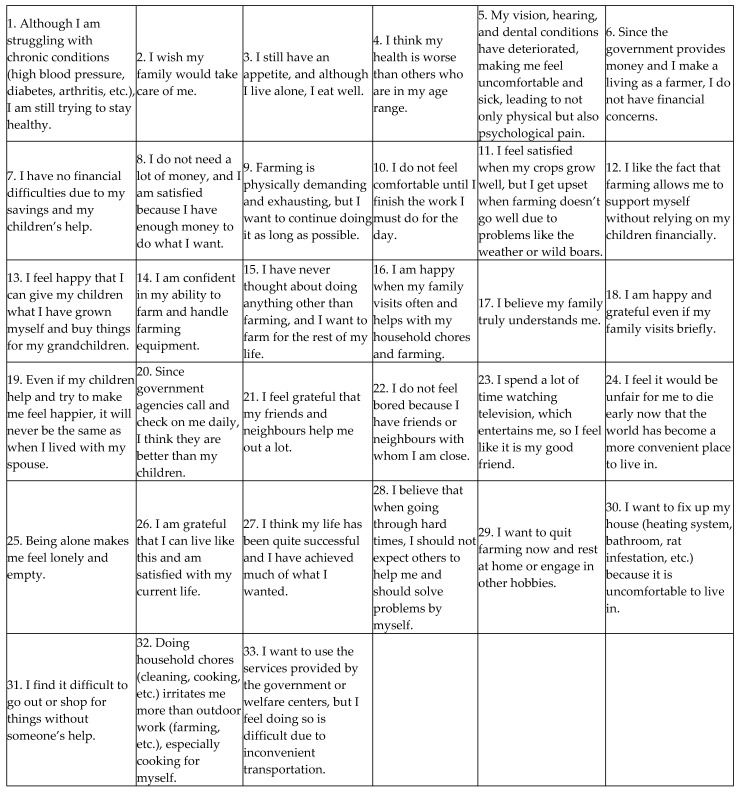
Q statements.

**Figure 3 behavsci-14-01150-f003:**
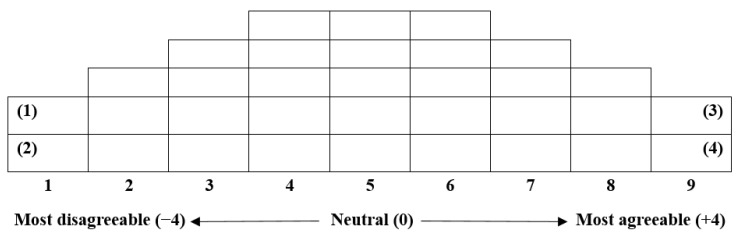
Q sorting grid.

**Figure 4 behavsci-14-01150-f004:**
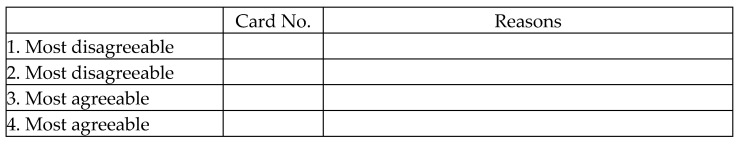
Q sorting sheet including reasons.

**Table 1 behavsci-14-01150-t001:** Interview subjects.

No.	Gender	Age	Region
1	Female	84	Gyeongsang-do Hapcheon
2	Female	74	Gyeongsang-do Hapcheon
3	Female	88	Gyeongsang-do Hapcheon
4	Female	84	Gyeongsang-do Hapcheon
5	Female	89	Gyeongsang-do Hapcheon
6	Male	75	Chungcheong-do Seosan
7	Male	82	Chungcheong-do Seosan

**Table 2 behavsci-14-01150-t002:** Q sample.

No.	Q Statements (Q Items)
Q1	Although I am struggling with chronic conditions (high blood pressure, diabetes, arthritis, etc.), I am still trying to stay healthy.
Q2	I wish my family would take care of me.
Q3	I still have an appetite, and although I live alone, I eat well.
Q4	I think my health is worse than others who are in my age range.
Q5	My vision, hearing, and dental conditions have deteriorated, making me feel uncomfortable and sick, leading to not only physical but also psychological pain.
Q6	Since the government provides money and I make a living as a farmer, I do not have financial concerns.
Q7	I have no financial difficulties due to my savings and my children’s help.
Q8	I do not need a lot of money, and I am satisfied because I have enough money to do what I want.
Q9	Farming is physically demanding and exhausting, but I want to continue doing it as long as possible.
Q10	I do not feel comfortable until I finish the work I must do for the day.
Q11	I feel satisfied when my crops grow well, but I get upset when farming doesn’t go well due to problems like the weather or wild boars.
Q12	I like that farming allows me to support myself financially without relying on my children.
Q13	I feel happy that I can give my children what I have grown myself and buy things for my grandchildren.
Q14	I am confident in my ability to farm and handle farming equipment.
Q15	I have never thought about doing anything other than farming, and I want to farm for the rest of my life.
Q16	I am happy when my family visits often and helps with my household chores and farming.
Q17	I believe my family truly understands me.
Q18	I am happy and grateful even if my family visits briefly.
Q19	Even if my children help and try to make me feel happier, it will never be the same as when I lived with my spouse.
Q20	Since government agencies call and check on me daily, I think they are better than my children.
Q21	I feel grateful that my friends and neighbours help me out a lot.
Q22	I do not feel bored because I have friends and neighbours with whom I am close.
Q23	I spend a lot of time watching television, entertaining me, so I feel like it is my good friend.
Q24	I feel it would be unfair for me to die early now that the world has become a more convenient place in which to live.
Q25	Being alone makes me feel lonely and empty.
Q26	I am grateful that I can live like this and am satisfied with my current life.
Q27	I think my life has been quite successful and I have achieved much of what I wanted.
Q28	I believe that when going through hard times, I should not expect others to help me and should solve problems myself.
Q29	I want to quit farming now and rest at home, or engage in other hobbies.
Q30	I want to fix my house (heating system, bathroom, rat infestation, etc.) because it is uncomfortable to live in.
Q31	I find it difficult to go out or shop for things without someone’s help.
Q32	Doing household chores (cleaning, cooking, etc.) irritates me more than outdoor work (farming, etc.), especially cooking for myself.
Q33	I want to use the government’s services or welfare centers, but I feel doing so is difficult due to inconvenient transportation.

**Table 3 behavsci-14-01150-t003:** P sample.

Type	P	Gender	Age	Education	Farming	Living Alone	Health	Assistance	Marital Status	Factor Weight
1	1	Female	83	Elementary school	<10 years	<10 years	Healthy	Children	Bereaved	1.4469
	8	Female	79	Elementary school	≥40 years	<10 years	Average	None	Bereaved	1.6730
	9	Female	70	Elementary school	<40 years	<20 years	Healthy	None	Bereaved	0.9847
	16	Female	84	Elementary school	≥40 years	<10 years	Very bad	Children	Bereaved	1.5749
	20	Female	81	No formal education	≥40 years	<40 years	Average	None	Bereaved	1.3076
2	3	Female	85	Elementary school	≥40 years	<20 years	Very bad	Children	Bereaved	0.8813
	4	Female	82	Elementary school	≥40 years	<20 years	Not good	Government	Bereaved	2.3504
	7	Female	81	No formal education	≥40 years	<30 years	Very bad	Children	Bereaved	0.7699
	10	Female	83	Middle school	<10 years	<20 years	Average	Children	Bereaved	1.4958
	11	Female	82	No formal education	≥40 years	<20 years	Not good	None	Bereaved	0.8219
	12	Female	76	Elementary school	<5 years	<30 years	Not good	None	Bereaved	0.4767
	13	Female	88	No formal education	≥40 years	<30 years	Very bad	None	Bereaved	0.7552
	15	Female	77	Elementary school	≥40 years	<20 years	Not good	None	Bereaved	0.8263
	18	Female	81	Elementary school	≥40 years	≥40 years	Average	Children	Bereaved	1.0638
	19	Female	90	Elementary school	≥40 years	<10 years	Average	Children	Bereaved	1.1766
	21	Female	83	Elementary school	≥40 years	<30 years	Average	None	Bereaved	0.1056
	24	Female	80	Middle school	<20 years	<20 years	Not good	Children	Divorced	0.3672
3	5	Male	83	Elementary school	<30 years	<10 years	Not good	Children	Bereaved	2.1036
	6	Female	86	Elementary school	<5 years	<20 years	Not good	Children	Bereaved	0.8572
	22	Male	80	No formal education	≥40 years	<30 years	Healthy	Children	Bereaved	1.1777
4	2	Female	68	Elementary school	≥40 years	≥40 years	Healthy	Children	Divorced	1.0834
	14	Female	88	No formal education	<40 years	<40 years	Healthy	None	Bereaved	1.4141
	17	Female	75	Elementary school	<20 years	<1 year	Not good	Community	Bereaved	0.7430
	23	Female	67	High school	<10 years	<10 years	Average	None	Bereaved	1.4631
	25	Female	81	Elementary school	<10 years	≥40 years	Average	None	Bereaved	0.5695

**Table 4 behavsci-14-01150-t004:** Eigenvalues and explanatory variances in the classification of the four types.

	Type			
Content	1	2	3	4
Eigenvalues	5.29	3.40	2.04	1.70
Variance (%)	21.14%	13.59%	8.14%	6.79%
Cumulative (%)	21.14%	34.73%	42.88%	49.67%

**Table 5 behavsci-14-01150-t005:** Correlations.

Type	1	2	3	4
1	1.000			
2	0.131	1.000		
3	0.090	0.172	1.000	
4	0.470	0.262	0.039	1.000

**Table 6 behavsci-14-01150-t006:** Type 1’s Statements and Z-Scores (≥1.00).

No.	Statements	Z Score
17	I believe my family truly understands me.	2.00
23	I spend a lot of time watching television, which entertains me, so I feel like it is my good friend.	1.84
2	I wish my family would take care of me.	1.62
1	Although I am struggling with chronic conditions (high blood pressure, diabetes, arthritis, etc.), I am still trying to stay healthy.	1.31
15	I have never thought about doing anything other than farming, and I want to farm for the rest of my life.	1.21
3	I still have an appetite, and although I live alone, I eat well.	1.19
16	I am happy when my family visits often and helps with my household chores and farming.	1.09
13	I feel happy that I can give my children what I have grown myself and buy things for my grandchildren.	1.02
20	Since government agencies call and check on me daily, I think they are better than my children.	−1.31
24	I feel it would be unfair for me to die early now that the world has become a more convenient place to live in.	−1.46
4	I think my health is worse than others who are in my age range.	−1.46
25	Being alone makes me feel lonely and empty.	−1.63

**Table 7 behavsci-14-01150-t007:** Type 2’s Statements and Z-Scores (≥1.00).

No.	Statements	Z Score
19	Even if my children help and try to make me feel happier, it will never be the same as when I lived with my spouse.	1.81
25	Being alone makes me feel lonely and empty.	1.48
18	I am happy and grateful even if my family visits briefly.	1.21
30	I want to fix my house (heating system, bathroom, rat infestation, etc.) because it is uncomfortable to live in.	1.18
26	I am grateful that I can live like this and am satisfied with my current life.	1.04
28	I believe that when going through hard times, I should not expect others to help me and should solve problems myself.	1.00
2	I wish my family would take care of me.	−1.05
27	I think my life has been quite successful, and I have achieved much of what I wanted.	−1.10
7	I have no financial difficulties due to my savings and my children’s help.	−1.73
8	I do not need a lot of money, and I am satisfied because I have enough money to do what I want.	−1.83
14	I am confident in my ability to farm and handle farming equipment.	−2.24

**Table 8 behavsci-14-01150-t008:** Type 3’s Statements and Z-Scores (≥1.00).

No.	Statements	Z Score
18	I am happy and grateful even if my family visits briefly.	1.72
25	Being alone makes me feel lonely and empty.	1.59
2	I wish my family would take care of me.	1.54
5	My vision, hearing, and dental condition have deteriorated, making me feel uncomfortable and sick, leading to not only physical but also psychological pain.	1.52
22	I do not feel bored because I have friends or neighbours with whom I am close.	1.25
7	I have no financial difficulties due to my savings and my children’s help.	1.23
20	Since government agencies call and check on me daily, I think they are better than my children.	1.14
1	Although I am struggling with chronic conditions (high blood pressure, diabetes, arthritis, etc.), I am still trying to stay healthy.	1.09
19	Even if my children help and try to make me feel happier, it will never be the same as when I lived with my spouse.	−1.15
3	I still have an appetite, and although I live alone, I eat well.	−1.15
14	I am confident in my ability to farm and handle farming equipment.	−1.34
8	I do not need a lot of money, and I am satisfied because I have enough money to do what I want.	−1.59
10	I do not feel comfortable until I finish the work I must do for the day.	−1.72
9	Farming is physically demanding and exhausting, but I want to continue doing it as long as possible.	−1.89

**Table 9 behavsci-14-01150-t009:** Type 4’s statements and Z-Scores (≥1.00).

No.	Statements	Z Score
13	I feel happy that I can give my children what I have grown myself and buy things for my grandchildren.	2.21
17	I believe my family understands me.	1.71
21	I feel grateful that my friends and neighbours help me a lot.	1.46
11	I feel satisfied when my crops grow well, but I get upset when farming doesn’t go well due to problems like the weather or wild boars.	1.25
18	I am happy and grateful even if my family visits briefly.	1.21
16	I am happy when my family visits often and helps with my household chores and farming.	1.07
3	I still have an appetite, and although I live alone, I eat well.	1.04
2	I wish my family would take care of me.	−1.10
5	My vision, hearing, and dental condition have deteriorated, making me feel uncomfortable and sick, leading to not only physical but also psychological pain.	−1.33
29	I want to quit farming now, and rest at home or engage in other hobbies.	−1.50
32	Doing household chores (cleaning, cooking, etc.) irritates me more than outdoor work (farming, etc.), especially cooking for myself.	−1.97

**Table 10 behavsci-14-01150-t010:** Statements consistent across the four types.

No.	Statement	Z Score
18	I am happy and grateful even if my family visits briefly.	1.24
1	Although I am struggling with chronic conditions (high blood pressure, diabetes, arthritis, etc.), I am still trying to stay healthy.	0.87
22	I do not feel bored because I have friends or neighbours with whom I am close.	0.77
31	I find it difficult to go out or shop for things without someone’s help.	−0.48
33	I want to use the services provided by the government or welfare centers, but I feel it is difficult due to inconvenient transportation.	−0.57
27	I think my life has been quite successful and I have achieved much of what I wanted.	−0.60

## Data Availability

The datasets generated and/or analysed during the current study are available from the corresponding author upon reasonable request.

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
