# Peer review of "Subjective Perception Types of Older Adults Living Alone as Farmers in Korea: A Q Methodology Study"

_behavsci, 2024, doi:10.3390/bs14121150_

Round 1

Reviewer 1 Report

Comments and Suggestions for Authors

Dear Author,

Your research is valuable in improving the mental health of elderly Korean farmers who live alone. I also think that your classification of their psychological characteristics using the Q method is highly original.

I have some comments and recommendations:

-Participants

The proportion of male participants was very low, and as a result, both of them belonged to type 3. Is it possible that women and men have fundamentally different subjective perceptions of living alone in rural areas? We speculated that gender differences may lead to differences in attitudes toward life, values, interactions with others in the community, economic situations, and support from those around them. Discussing the results of the analysis limited to women may clarify the focus of the research.

- Naming of the factor

Regarding Type 2, "Independent Life and Change Pursuit," looking at the results of the Q principal component factor analysis, I could not find any Q-Sorting sentences that mean "Change Pursuit." I thought it might be more appropriate to limit it to "Independent Life." Please tell me why you named it "Change Pursuit."

-Discussion

As mentioned above, I don't think type 2 is meaning as "the type that does not settle in the given environment but seeks a new life." Rather than seeking a new life, I think that they place more importance on solving problems by themselves without relying on others.

-Conclusion

1.In the conclusion stage, you use the framework of ‘relationships,’ ‘work (farming),’ and ‘emotions’ to show the four types of relationships and structures. In the results, you do not discuss the explanation for this structure diagram, which makes it difficult to understand this structure. If the research revolves around three main themes, you should provide a logical explanation for the relationships between the three themes and the four types. Also, if you show a structure diagram, it would be better to clarify the meaning of the arrangement of each type.

If the three themes are not important in this research, I think it would be more logically consistent to discuss the main research results using only the four types.

2. In the last paragraph, it is stated that type 1 and type 3 require practical care. The meaning of this "practical care" is unclear. I thought that the meaning of "practical care" may be different between type 1 and type 3. I recommend that you express it more specifically.

Author Response

Your research is valuable in improving the mental health of elderly Korean farmers who live alone. I also think that your classification of their psychological characteristics using the Q method is highly original.

I have some comments and recommendations:

  1. Participants

The proportion of male participants was very low, and as a result, both of them belonged to type 3. Is it possible that women and men have fundamentally different subjective perceptions of living alone in rural areas? We speculated that gender differences may lead to differences in attitudes toward life, values, interactions with others in the community, economic situations, and support from those around them. Discussing the results of the analysis limited to women may clarify the focus of the research.

Response: Thank you for your feedback.

When selecting a P-sample in Q methodology, it is sufficient to choose individuals related to the research topic, but this does not necessarily require following the statistical representation of demographic backgrounds. In the research results, demographic characteristics can help identify the features of each type. Therefore, even if there are only two male participants, the results including their data are still meaningful because in Q methodology, the focus is not to find a generalization based on demographic characterisitics, but to see which type each individual for this particular study falls under. A quantitative research can be done focusing on the statistical representation of demographic backgrounds. This point has been addressed in the limitations section.

  1. Naming of the factor

Regarding Type 2, “Independent Life and Change Pursuit,” looking at the results of the Q principal component factor analysis, I could not find any Q-Sorting sentences that mean “Change Pursuit.” I thought it might be more appropriate to limit it to “Independent Life.” Please tell me why you named it “Change Pursuit.”

Response: Thank you for your valuable feedback.

I want to explain the rationale behind naming Type 2 as “Independent Life and Change Pursuit. ” Participants in this group highly value maintaining an independent life while caring for themselves. However, we also found that they tend to introduce changes to their current lives to enhance their quality of life. For instance, they wanted to improve their living environment, meet friends, and engage in hobbies. Participants in this type have shared sentiments such as, “I want to enjoy this wonderful world by meeting friends and pursuing hobbies,” “I want to live longer,” and “I’d like to build or renovate a new home for a better life.” These expressions suggest that they are not content with the current state but are striving for a better life, so we initially included the concept of “pursuing change” in the naming.

However, your comments provided us with new insight. These individuals may not necessarily seek an entirely new life but aim to improve their existing circumstances. Considering this, we revised the name of Type 2 to “Independent Life and Improvement-Seeking.” This revised title better captures their independent attitude of resolving issues independently while expressing a desire to enhance their current environment. The term “improvement” seems more suitable than “change,” as it better reflects their tendency to focus on elevating the quality of life within their current situation rather than pursuing a completely new one.

  1. Discussion

As mentioned above, I don’t think type 2 is meaning as “the type that does not settle in the given environment but seeks a new life.” Rather than seeking a new life, I think that they place more importance on solving problems by themselves without relying on others.

Response: In response to the reviewer’s feedback, we revised the naming of Type 2 and made corresponding adjustments to the discussion section. Thank you for your insightful comments, which have helped clarify our understanding of this group’s motivations.

  1. Conclusion

4-1 .In the conclusion stage, you use the framework of ‘relationships,’ ‘work (farming),’ and ‘emotions’ to show the four types of relationships and structures. In the results, you do not discuss the explanation for this structure diagram, which makes it difficult to understand this structure. If the research revolves around three main themes, you should provide a logical explanation for the relationships between the three themes and the four types. Also, if you show a structure diagram, it would be better to clarify the meaning of the arrangement of each type.

If the three themes are not important in this research, I think it would be more logically consistent to discuss the main research results using only the four types.

Response: Thank you for your valuable feedback. We reviewed the relationships between the three structural diagrams and the four types again. As a result, we concluded that the three diagrams could create confusion in interpreting the findings rather than adequately explaining the four types. Following your suggestion, we decided to focus solely on the four types in discussing the research results, which ultimately enhances the logical consistency of the study. By concentrating on these four types, we clarified the focus of the research, making it easier for readers to understand the findings. Once again, we appreciate your insightful feedback.

4-2. In the last paragraph, it is stated that type 1 and type 3 require practical care. The meaning of this “practical care” is unclear. I thought that the meaning of “practical care” may be different between type 1 and type 3. I recommend that you express it more specifically.

Response: Per your suggestion, we revised the conclusion section to clarify the term “practical care,” which was somewhat ambiguous. We have now provided a more detailed explanation to differentiate between Types 1 and 3. Thank you for your valuable input, which has helped enhance the clarity of our findings.

Reviewer 2 Report

Comments and Suggestions for Authors

The article entitled “Subjective Perception Types Regarding the Lives of Elderly Farmers Living Alone in Korea” was considered for review. The authors aimed "to explore the subjective perception types of elderly farmers living alone in Korea”. The article is relevant, has scientific merit, and is well-written. The solidity of the method is highlighted. Despite this, some recommendations are indicated:

I suggest clarifying which author (authors) conducted the interview, as well as what the researcher's credentials were, for example, PhD, medical doctor, psychologist. In addition, I recommend explaining whether the researcher was trained and what type of relationship, if any, was established before the study began.

Specifically in the Abstract, I recommend indicating important aspects of the method that were not mentioned, such as: place and date of data collection, number of participants in each phase of the study. In addition, the objective presented in the Abstract is different from that presented in the Introduction section. It would be beneficial to explain the reason for this discrepancy, which may require adaptation.

In the Materials and Methods section, it is essential to clarify how the literature review was conducted, whether, for example, descriptors or keywords were used to conduct the searches, and, if possible, determine the databases that were used. In addition, it is not clear how the data were extracted and analyzed so that the items could be presented as statements. I also recommend presenting some clarifications in item 2.3. (P-Sample Selection), namely: how were the participants selected? For example, convenience, consecutive sampling, and snowball: How were the participants approached? For example, in person, by telephone, letter, or e-mail, how many participants were approached to participate in the study, and how many people refused to participate or withdrew? For what reasons? Where was the interview conducted? For example, at the elderly person's home or clinic. Was there anyone else besides the participants and researchers at the time of data collection? How long did data collection last in this phase of the research?

The Results are presented clearly and in a logical sequence. The objective highlights the main results. The tables contain helpful information, are numbered, arranged appropriately, and are self-explanatory, with no overlapping data. I suggest reviewing the layout of Figure 2 for readability and clarity of the information presented.

The Discussion is well written, presenting similarities about other authors but emphasizing the new aspects of the study. The author's interpretations show security and propriety, advancing knowledge and allowing its reproduction. I suggest mentioning possible generalizations and practical applications more consistently. The limitations were also presented satisfactorily. The consistency between the data presented, results, and interpretation performed stands out, with advances in relation to what has already been produced in the literature, despite the limitations related to the external validity of the findings. The Conclusions are clear, respond to the objectives, and are based on the study's findings. The References are adequate but need to be updated since 50% are from before the last five years.

Author Response

The article entitled “Subjective Perception Types Regarding the Lives of Elderly Farmers Living Alone in Korea” was considered for review. The authors aimed “to explore the subjective perception types of elderly farmers living alone in Korea”. The article is relevant, has scientific merit, and is well-written. The solidity of the method is highlighted. Despite this, some recommendations are indicated:

1.I suggest clarifying which author (authors) conducted the interview, as well as what the researcher’s credentials were, for example, PhD, medical doctor, psychologist. In addition, I recommend explaining whether the researcher was trained and what type of relationship, if any, was established before the study began.

Response: Thank you for your feedback. In response to the reviewer’s comments, we have provided specific details about the researchers who conducted the interviews in the Materials and Methods section. Additionally, we included information on the training these researchers received for this study, along with a description of the relationships established with stakeholders while seeking cooperation from the research participants. Thank you for your constructive feedback, which has helped improve the clarity and comprehensiveness of our manuscript.

2.Specifically in the Abstract, I recommend indicating important aspects of the method that were not mentioned, such as: place and date of data collection, number of participants in each phase of the study. In addition, the objective presented in the Abstract is different from that presented in the Introduction section. It would be beneficial to explain the reason for this discrepancy, which may require adaptation.

Response: Thank you for your feedback. We revised the abstract accordingly. Additionally, we also added these details in the Materials and Methods section.

We hope these revisions meet the reviewer’s expectations.

3.In the Materials and Methods section, it is essential to clarify how the literature review was conducted, whether, for example, descriptors or keywords were used to conduct the searches, and, if possible, determine the databases that were used. In addition, it is not clear how the data were extracted and analyzed so that the items could be presented as statements.

Response: Thank you for your feedback. We specified the keywords, databases, and literature review methods used in our search. Additionally, we elaborated on how we selected the Q sample from the Q population as extracted from the literature. Your suggestions have greatly improved the clarity and rigor of our methodology.

4.I also recommend presenting some clarifications in item 2.3. (P-Sample Selection), namely: how were the participants selected? For example, convenience, consecutive sampling, and snowball: How were the participants approached? For example, in person, by telephone, letter, or e-mail, how many participants were approached to participate in the study, and how many people refused to participate or withdrew? For what reasons? Where was the interview conducted? For example, at the elderly person’s home or clinic. Was there anyone else besides the participants and researchers at the time of data collection? How long did data collection last in this phase of the research?

Response: Thank you for your valuable feedback. We have revised as much as we could that was relevant to our study as advised in the Materials and Methods section. Your suggestions have significantly enhanced the clarity and detail of our methodology.

The Results are presented clearly and in a logical sequence. The objective highlights the main results. The tables contain helpful information, are numbered, arranged appropriately, and are self-explanatory, with no overlapping data. I suggest reviewing the layout of Figure 2 for readability and clarity of the information presented.

Response: Thank you for your valuable feedback. We added a detailed explanation of Figure 2 to enhance the readability and clarity of the information presented. We hope these revisions meet the reviewer’s expectations.

The Discussion is well written, presenting similarities about other authors but emphasizing the new aspects of the study. The author’s interpretations show security and propriety, advancing knowledge and allowing its reproduction. I suggest mentioning possible generalizations and practical applications more consistently. The limitations were also presented satisfactorily. The consistency between the data presented, results, and interpretation performed stands out, with advances in relation to what has already been produced in the literature, despite the limitations related to the external validity of the findings. The Conclusions are clear, respond to the objectives, and are based on the study’s findings. The References are adequate but need to be updated since 50% are from before the last five years.

Response: Thank you so much. We made major revisions to enhance the consistency of possible generalizations and practical applications. Additionally, we updated the references to include more recent contents.